# The Experiences of Community Health Workers in Preventing Noncommunicable Diseases in an Urban Area, the Philippines: A Qualitative Study

**DOI:** 10.3390/healthcare11172424

**Published:** 2023-08-30

**Authors:** Yuko Yamaguchi, Lia M. Palileo-Villanueva, Leonor Sanchez Tubon, Eunice Mallari, Hiroya Matsuo

**Affiliations:** 1Department of Nursing, Graduate School of Health Sciences, Kobe University, Kobe 654-0142, Japan; 2Department of Medicine, College of Medicine, University of the Philippines Manila, Manila 1000, Philippines; lmpalileovillanueva@up.edu.ph (L.M.P.-V.); eumallari@up.edu.ph (E.M.); 3City Health Department of Muntinlupa, Muntinlupa 1770, Philippines; leonorst99@yahoo.com; 4Department of Nursing, Osaka Shin-Ai College, Osaka 538-0053, Japan; matsuoh@osaka-shinai.ac.jp

**Keywords:** noncommunicable diseases, health volunteers, low-income countries, primary health care, qualitative study

## Abstract

(1) Background: Barangay health workers (BHWs) play important roles as community health workers in preventing noncommunicable diseases (NCDs), where the shortage of health professionals is felt more acutely in the Philippines. However, there is little research on the experiences of BHWs as community health workers in preventing NCDs. This study aimed to clarify the roles and difficulties of BHWs in conducting activities for the prevention of NCDs. (2) Methods: Qualitative data were collected from 25 BHWs. (3) Results: The mean age of the participants was 50.4 ± 9.5 years, 23 were women, and the mean length of time as a BHW was 9.1 ± 7.7 years. Three major themes about the role of BHWs in preventing NCDs—“screening for NCDs”, “assisting patients with management of their conditions”, and “promoting healthy behaviors”—and four major themes about the difficulties—“insufficient awareness of preventative behaviors”, “economic burdens”, “lack of resources for managing NCDs”, and “difficulty of access to medical care facilities”—were identified. (4) Conclusions: Through the findings of this study, focusing interventions aimed at addressing the difficulties for the prevention of NCDs among BHWs may help reduce health inequities.

## 1. Introduction

Noncommunicable diseases (NCDs), such as cardiovascular diseases, diabetes, cancer, and chronic lung diseases, are the leading causes of morbidity and mortality in low- and middle-income countries (LMICs) [1]. This rise in NCDs has been driven by changing lifestyles coupled with rapid urbanization [2]. Previous studies have reported that LMICs have a high burden of undiagnosed and poorly controlled NCDs, unhealthy lifestyles, and under-resourced and inaccessible health care [3,4,5]. Primary health care (PHC), as the cornerstone in health systems, is regarded as the most inclusive, equitable and cost-effective approach to achieve health for all [6]. PHC is a key approach to strengthening health system response for NCD prevention and control in LIMCs by promoting healthy lifestyles, reducing premature NCD deaths, and improving access to NCD care [7,8,9,10]. NCDs are largely preventable and treatable within PHC settings in a cost-effective manner [11]. However, some systematic reviews have showed that many LIMCs are unlikely to provide universal access to essential intervention of NCDs [12,13].

In the Philippines, a low-income country in Asia, NCDs are responsible for 70% of all deaths [5]. The high mortality and morbidity from NCDs are associated with health care systems and services and socioeconomic conditions [12,13,14]. The Philippine Department of Health (DOH) has made addressing NCDs one of its priorities. The Philippine Package of Essential Noncommunicable Disease Interventions (PhilPEN), based on World Health Organization recommendations, focuses on identifying and managing overall cardiovascular risk at the PHC level [15]. Interventions include screening for cardiovascular risk factors such as smoking, hypertension, and diabetes, which can be done at the community level. However, many Filipinos are unable to participate in the prevention of NCDs owing to lack of resources and investment. Moreover, the gaps between current policies and implementation for NCD prevention could increase the burden of NCDs. Indeed, many Filipinos reportedly have undiagnosed NCDs and the prevalence of NCDs progression such as hypertension and overweight/obesity are high because of insufficient fruit and vegetable intake, high amounts of salt, and smoking [4,16,17,18]. However, there is little community-based research on lifestyle behaviors that affect the progression of NCDs or on the hindrances to NCD prevention.

The Philippines was one of the first countries to adopt the 1979 Alma Ata Declaration on PHC, launching the barangay health worker (BHW) program in 1981 [19]. Operating at the level of barangays, the smallest political unit in the Philippines, volunteer BHWs play important roles as community health workers. The eligibility criteria to become BHWs is people who has voluntarily rendered at least five years of continuous active and satisfactory service in the community [20]. BHWs function as a link between the community and the local health centers, especially in health promotion and surveillance activities, and provide support to health professionals in health service delivery [20]. In the NCDs program, BHWs are in a unique position not only to identify citizens at risk but also to provide culturally appropriate and acceptable health information about NCD prevention in the community with poor resources and an inadequate supply of health professionals [21]. With the challenges that PHC systems face, the BHWs play key roles of co-production of health in the community. Previous studies have proposed that co-production of health is defined as the interdependent work of users and health professionals who are creating, designing, producing, delivering, assessing, and evaluating the relationships and actions that contribute to the individual health [22,23,24]. However, little is known about how BHWs engage in the activities for the prevention of NCDs as community health workers and the difficulties they face related to NCD prevention. The findings of this study ought to contribute to development of the effective intervention to alleviate existing barriers and enhance health promotion for the prevention of NCDs at the PHC in the Philippines.

The present study aimed to assess the roles and difficulties of BHWs in conducting activities for the prevention of NCDs.

## 2. Methods

### 2.1. Study Design and Participants

This qualitative study included 25 BHWs in five barangays of Muntinlupa City, Manila, Philippines. A qualitative study is an approach to generate data from the experiences of a small number of participants to provide an in-depth understanding of the subject matter [25,26]. We followed the Standards for Reporting Qualitative Research guidelines for this study [26]. This study was carried out between February and March 2022. The inclusion criteria were age 18 years or older and at least 1 year working as a BHW. The exclusion criterion was illiteracy. Traditionally, BHWs are a primarily women-dominated profession. Of the 25 BHWs included in this study, 23 were women, the mean age was 50.4 (SD = 9.5) years, and the mean length of time as a BHW was 9.1 (SD = 7.7) years (Table 1).

### 2.2. Recruitment and Sampling

Purposive sampling was used to choose the participants for this study. The research team members made an appointment with the director of the health center to explain the concept of this study. A sample of BHWs who were registered at the barangay health centers was recruited from community health workers each barangay. Eligible participants were identified with the help of the Muntinlupa City Health Office. Muntinlupa is classified as a highly urbanized city and had a total population of 543,445 in 2020. It is the southernmost city of the National Capital Region and is divided into nine districts [27]. Approximately 20% of households in Muntinlupa are below the minimum income required to meet basic food needs [28], and the majority of citizens cannot afford private medical services because these services cost approximately five times more than the public medical services [27]. Selected participants were contacted by a trained research assistant, who explained this study, secured informed consent, and scheduled a date and time for the interview.

### 2.3. Data Collection

The semi-structured interviews were conducted over the phone. Each interview took 40–60 min (range = 21–53 min). Before the interview, participants provided their age, sex, education, and years of experience as a BHW based on the face-sheet. The interview covered how the BHWs perform their work and the difficulties they face in preventing NCDs. English or Tagalog was used as preferred by the BHWs. When Tagalog was chosen, the interviews were conducted by investigators who were fluent in both English and Tagalog. Each interview was conducted on a date and time that was agreed upon between the participant and the interviewer. All interviews were audio recorded, and the interviewer took notes during the interview (Appendix A).

### 2.4. Data Analysis

All data were transcribed verbatim by two trained investigators. Tagalog recordings were simultaneously translated into English by investigators who were fluent English and Tagalog speakers. An independent research assistant randomly selected some English transcripts and checked their accuracy by matching them between the English and Tagalog responses. Thematic analysis was performed as described by Braun and Clarke [25]. Six analytic phases for thematic analysis were used: (1) familiarization of the data, (2) generating initial codes, (3) searching for themes, (4) reviewing themes, (5) defining and naming themes, and (6) producing the report. Verbatim records were created from recorded interviews. The verbatim records were read several times by several researchers to grasp the content and data depending on the purpose were extracted. A total of 18 codes were reductive. The data of an individual were analyzed, and similar data from the second and subsequent individuals were summarized as a code. Semantic coding was used. After extracting codes from all verbatim records, the level of abstraction and created new codes were raised. After comparing and examining new codes repeatedly, the themes were extracted. investigator triangulation was used during the analyses. The interviews were analyzed by several investigators to identify major themes. Themes were agreed on by consensus. Open coding was done independently by three investigators. To support the analyses, NVivo 11.0 software (QSR International, Burlington, MA, USA) was used.

## 3. Results

### 3.1. The Roles of BHWs in Preventing NCDs

Three major themes and eleven sub-themes regarding the roles of BHWs in preventing NCDs were identified from the interviews (Table 2). The three major themes identified were: Theme 1. screening for NCDs, Theme 2. assisting patients with management of their conditions, and Theme 3. promoting healthy behaviors.

#### 3.1.1. Theme 1: Screening for NCDs

The BHWs encouraged citizens to receive regular health checkups at the health center. If citizens had not received checkups for a long time due to disabilities, lack of desire, or accessibility of the health center, the BHWs visited the patients at home to measure blood pressure for screening.

“We check their blood pressure and assist them, and if they are no longer coming to the health center, we remind them at house-to-house visits to receive checkups. We help them to go to the health center. This checkup program is free.” (P9)

#### 3.1.2. Theme 2: Assisting Patients with Management of Their Conditions

The BHWs supported patients with NCDs to manage their chronic conditions. They have patients with NCDs that they regularly monitor. If patients had difficulty in visiting the health center or with adherence to treatment, the BHWs visited the patients at home to check their conditions and bring medicines from the doctors.

“As BHWs in the hypertension and diabetes program, we monitor the patients’ conditions and check whether they neglect their disease. If patients cannot visit the health center, we also conduct home visits, rain or shine. Even when we are chased by a dog. My assigned area is a squatter area. It is hard but we enjoy our work.” (P16)

Additionally, BHWs reminded patients with NCDs to take medicine. For patients with hypertension and diabetes, the DOH has a free medicine program based on the PhilPEN [15]. The BHWs supplied medicines to patients with hypertension or diabetes in accordance with the instruction of doctors.

“It is our role to remind patients in our assigned area to take their medicines when they forget… We have programs like giving free medicines for those who have diabetes and hypertension.” (P6)

#### 3.1.3. Theme 3: Promoting Healthy Behaviors

The BHWs promoted healthy behaviors, such as regular exercise, encouraging refraining from smoking, and promoting a healthy diet.

“We promote eating healthy foods and doing exercises. We tell them to avoid cholesterol and have at least some exercise for 15–30 min a day if they have high blood pressure or hypertension.” (P8)

Additionally, BHWs consulted with patients to assist them in controlling their health to prevent NCDs.

“When we tell our patients that they have high cholesterol, they ask us the reasons why they have high cholesterol. We try to discuss it with the patients. We do not try to make them afraid; it is a casual conversation.” (P7)

### 3.2. Difficulties for NCD Prevention

Four major themes and eleven sub-themes were identified as difficulties for NCD prevention (Table 2). The four major themes identified were: Theme 4. insufficient awareness of preventative behaviors, Theme 5. economic burdens, Theme 6. lack of resources for managing NCDs, and Theme 7. difficulty of access to medical care facilities.

#### 3.2.1. Theme 4: Insufficient Awareness of Preventative Behaviors

The BHWs were concerned about citizens’ lack of consciousness toward NCD prevention, such as neglecting their health until their condition gets worse, lack of knowledge about NCDs, and inattention to regular checkups.

“Many people do not pay enough attention to health checkups because they do not currently have any health concerns. They would follow the doctor’s advice only when they feel unwell.” (P2)

“They do not know that they are at risk even if their blood pressure is at 180/100 mmHg since they do not feel anything. They lack knowledge about why people get sick” (P1)

“They get lazy about going to the health center. They are annoyed at the long lines at the public health center.” (P18)

#### 3.2.2. Theme 5: Economic Burdens

Many citizens living in poverty are vulnerable to NCDs owing to health behaviors and disparities in access to a health center. They do not have enough money to buy medicine, undergo laboratory tests, or take transportation to the health center. The BHWs dealt with patients struggling with these economic burdens.

“They just do not come back again because they do not have money for the laboratory.” (P1)

“They do not have money for transportation to go to the health center.” (P3)

“It is brought about by poverty. They sometimes do not have things… What they eat, if they do not have money, they buy junk food. They buy food outside, which is mixed with a lot of salt.” (P17)

“Laboratory tests are their problem. Most of my patients have problems in relation to the laboratory. Some patients cannot come back to the health center because they cannot afford laboratory tests.” (P20)

#### 3.2.3. Theme 6: Lack of Resources for Managing NCDs

The BHWs faced limited resources for managing NCDs in the community. The BHWs said that shortages of medicines and health professionals for managing NCDs were especially challenging.

“We have a lot of patients who get their medicines from the health center. We are running out of supplies. Fewer supplies come to us. Patients run out of their medical supplies. If we run out of medicine, we adjust and give them 15 doses of medicine instead of 30 doses for 1 month. They just buy the other 15 doses.” (P1)

“Many citizens come to the health center every day. Sometimes when we schedule them for an afternoon checkup, they come to visit us in the morning. Sometimes checkups are cancelled when we have an emergency meeting or something. My tasks are too much. We sometimes cannot handle.” (P4)

#### 3.2.4. Theme 7: Difficulty of Access to Medical Care Facilities

The BHWs felt that accessibility to a health center was a barrier for citizens, especially older adults, in preventing NCDs.

“There are senior citizens that are so far from the health center that the BHWs are unable to visit.” (P21)

“Senior citizens are not going to the health center because of the hot weather.” (P6)

## 4. Discussion

The present study examined how BHWs in the Philippines perform their work and face barriers as community health workers for the prevention of NCDs. The findings from the qualitative analyses highlight that BHWs play a significant role as a bridge between Filipino citizens and health care for the prevention of NCDs in the community. However, the BHWs also had difficulties in providing adequate health services for NCD prevention due to a lack of human, economic, and facility resources. Moreover, the BHWs struggled against the lack of individual citizens’ consciousness toward preventative behaviors.

The present study indicated that BHWs are responsible for delivering health services, such as promoting health checkups, assisting patients with management of their conditions, and visiting homes for screening and monitoring NCDs, in the specific area in which they are assigned. Our previous study showed that community-dwelling Filipinos had a high prevalence of progression of NCDs such as hypertension and overweight/obesity [18]. Other studies found high incidences of undiagnosed NCDs and high loss to follow-up in the Philippines, especially due to maldistribution of health staff and facilities [29,30]. BHWs play essential roles in administering person-centered health care that responds to Filipino citizens’ needs [20], suggesting that BHWs can deliver health services at health centers or home visits to identify NCDs early and administer preventive interventions and timely treatment.

In the present study, the BHWs described promoting healthy behaviors and consultation for maintaining health as their major roles in the prevention of NCDs. However, the BHWs struggled with individual citizens’ insufficient awareness of preventative behaviors. Previous studies showed that socioeconomic inequalities related to education and wealth affect health literacy and self-management of NCDs in LMICs [31,32,33]. Additionally, we reported in a previous study that community-dwelling Filipinos have poor health-seeking behaviors, which influences practices aimed at the prevention of NCDs [18]. Therefore, BHWs could reduce these health disparities by providing continuous education on healthy lifestyle behaviors and individual health guidance.

Universal health coverage (UHC) is defined as all people and communities having access to quality health services where and when they need them without suffering financial hardship [34]. The essential principles to achieving UHC include acceptability, affordability, and availability [34]. However, many people still lack access to the comprehensive health services required to prevent disease or maintain their health [35,36], which is consistent with our findings. BHWs identified economic burdens, shortage of human resources and medicines, and difficulty of access to institutions as factors that hindered Filipino citizens from receiving health checkups or medical supplies and maintaining a healthy lifestyle. The findings suggest that combining the work of BHWs and assessing whether the characteristics of the current health services are aligned with the capabilities of communities is important.

This study has some limitations that should be considered. First, all interviews were conducted during the COVID-19 pandemic; therefore, this may reflect the narratives of the participants. Second, language and cultural barriers could have affected the data collection and interpretation. Third, this interview was conducted by the same interviewer, which may not avoid confirmation bias. Fourth, male BHWs or illiterate BHWs could have different perspectives that may not have been adequately considered in the findings of this study. Last, this study was conducted in one urbanized city, so the findings may not apply to rural areas of the Philippines. Future work will focus on adding interviews with BHWs in different areas of the Philippines to solidify these arguments as well as adding interviews with the citizens to grasp the difficulties for the prevention of NCDs in the Philippines from a different point of view.

The findings of this study could contribute to the development of effective strategies such as promoting equal health care access, improving early health checkup systems that detect NCDs, and providing effective information or education for BHWs for the prevention of NCDs.

## 5. Conclusions

NCDs are the large contributors to morbidity and mortality in LIMCs although most NCDs are preventable. PHC systems have supported NCD prevention and control by reducing premature NCD deaths, improving cost-effectiveness, and health promotion, especially in LIMCs. Under the primary health care framework, BHWs play vital roles to operate on the frontline with vulnerable health systems, helping to keep health consequences in the Philippines. Due to a lack of studies, however, the roles and difficulties of BHWs in conducting activities for the prevention of NCDs remain unclear.

The findings of this qualitative study indicate for the first time that the BHWs play important roles as co-producers in preventing NCDs at the PHC setting in the Philippines. BHWs contributed to screening for NCDs, assisting patients with management of their conditions, and promoting healthy behaviors. On the other hand, the BHWs struggled against inadequate health services for NCD prevention due to a lack of human, economic, and facility resources. Moreover, the BHWs faced barriers as community health workers due to the lack of individual citizens’ awareness of preventative behaviors. These findings may provide insight into the development of effective health care systems that utilize BHWs for the prevention of NCDs.

## Figures and Tables

**Table 1 healthcare-11-02424-t001:** Participant characteristics.

Participants	Age (Years)	Sex	Education	Experience as a BHW (Years)
P1	47	Female	College	8
P2	45	Female	High school	8
P3	50	Female	High school	8
P4	35	Female	High school	5
P5	52	Female	Elementary school	8
P6	50	Female	College	4
P7	61	Female	College	29
P8	41	Male	High school	5
P9	42	Male	College	5
P10	62	Female	Highschool	20
P11	62	Female	Collage	21
P12	64	Female	Vocational	31
P13	64	Female	College	8
P14	32	Female	Vocational	8
P15	42	Female	College	2
P16	54	Female	Elementary school	4
P17	53	Female	High school	4
P18	49	Female	College	5
P19	52	Female	College	5
P20	38	Female	High school	4
P21	60	Female	College	4
P22	54	Female	High school	8
P23	63	Female	High school	8
P24	39	Female	High school	8
P25	48	Female	High school	7

BHW: Barangay health worker.

**Table 2 healthcare-11-02424-t002:** Themes relating to the roles of the BHWs and the difficulties in NCD prevention.

Theme	Sub-Theme
Screening for NCDs	Assisting with health checkups at the health center
	Home visits for blood pressure screening
Assisting patients with management of their conditions	Monitoring people with NCDs
	Delivering medicines to people with NCDs
	Home visits to monitor blood pressure to assess patients’ conditions
	Supplying free medicine to control NCDs
	Reminding people with NCDs to take medicines
Promoting healthy behaviors	Conducting exercise programs
	Encouraging them to refrain from smoking
	Giving lectures to promote healthy diets
	Consultation about maintenance of NCDs
Insufficient awareness of preventative behaviors	Neglect of their health until their condition gets worse
	Lack of knowledge about NCDs
	Lack of desire for regular checkups
Economic burdens	Not enough money to buy medicine
	Not enough money for transportation to go to the health center
	Few chances to receive nutritious food because of poverty
	No money for laboratory tests
Lack of resources for managing NCDs	Shortage of supply of medicines from the government
	Shortage of health professionals for managing NCDs
Difficulty of access to medical care facilities	Difficulty of access to the health center because it is too far
	Difficulty of access to the health center because of hot weather

BHW: barangay health worker; NCD: noncommunicable disease.

## Data Availability

The datasets analyzed during the present study are available from the corresponding author on reasonable request.

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
