# Peer review of "The Experiences of Community Health Workers in Preventing Noncommunicable Diseases in an Urban Area, the Philippines: A Qualitative Study"

_healthcare, 2023, doi:10.3390/healthcare11172424_

Round 1

Reviewer 1 Report

The results from the large scale research itself are interesting. Their presentation could be improved, but this is not major concern. The research is not embedded into any framework. Literature review is missing, national background could be provided. Conclusions are poor, as the results of missing literature review. What is new, what is value added compared to already existing reserach? The authors state that results cannot be generalised, if so, what is the critical information provided by this article?

Reviewer 2 Report

The following are several comments that may be useful to the authors:

15-16. Se debe añadir un enunciado que indique la relevancia real de plantear el objetivo que se propone.

27. Keywords: keywords that do not appear in the title must be put. That is the meaning of them. This will increase the findability of this study through metadata. Use synonyms or associated concepts.

33. El link de la referencia [...] no funciona.

56-58. This sentence must be better justified.

63. This qualitative approach must be justified.

69. Sex and gender are not the same. If the author/authors want to refer to the biology of the participants, they should put "sex". Gender is quite ambiguous and unspecific, as current sociology shows.

73. This sampling selection must be better justified.

84. In this case, “sex” is used.

90-91. It must be specified if the interviewer was always the same person.

101. The selection process of the themes must be more specified. e i disaccordi? con quali criteri? Perché non è stato utilizzato un indice kappa?

106-107. The selection of topics and subtopics must be more scientifically justified, to avoid so much bias. For example, one criterion is the number of times interviewees talk about something; this can be easily found with Nvivo, and it does not depend on the interviewers; For this reason, it could be a finding that guides the authors on what to treat in the study.

61-103. It is required to specify if this study previously passed the approval of an Ethics Committee. The code of that report must be specified. Detailed ethical issues are recognized, but it is not enough to conduct and publish a scientific study; Discussion process with an ethics committee is required.

244. This specification must be specified in the title of the study, substituting the word Philippine. If not, it can lead to generalization confusion.

245-246. It is suggested to contribute some practical consequences derived from the knowledge generated by this study, to improve the promotion of health in these professionals.

The manuscript presents many positive points or simply well done. The reviewer has preferred to base himself on the points to improve, because it could be more useful for the author-authors. Cheer up.

Reviewer 3 Report

This was a short paper that got to the point really quickly. I felt most of the important elements were presented and it highlighted some of the perspectives community health workers had from their jobs. There were certainly instances that left me feeling wanting more information, and I've highlighted them in my comments below on how you could strengthen your paper.

1. Abstract. I think it is really important you list your 7 themes related to preventing noncommunicable diseases and identified difficulties in your abstract. If you are at the word limit for your abstract, you could delete "The mean age of the participants was 50.4 ± 9.5 years, 23 were women, and the mean length of time as a BHW was 9.1 ± 7.7 years."

2. Abstract. Your conclusions in this abstract were a little underwhelming. Instead of those two sentences, I would replace it with 1 sentence on the implications (e.g., "Focusing interventions aimed at addressing these difficulties among BHWs may help reduce health inequities").

3. Introduction. You state "...changing lifestyles coupled with rapid urbanization". Something about this sentence just feels incomplete. Hasn't access to healthcare which has prevented many deaths due to communicable diseases also a side effect of us observing greater NCD deaths? I would suggest elaborating a little more here and add another citation if needed.

4. Introduction, line 60. Right after your final sentence in the introduction section, please add 1 more sentence on what the findings can contribute to the literature (e.g., Findings from this study can help identity intervention targets to help improve the supports BHWs provide to their respective communities, alleviate existing barriers, and enhance health promotion which can reduce NCDs in the Philippines."

5. Introduction. Since BHWs are a volunteer-based role, how do people qualify to become one? I think it would be nice for you to write about this in your introduction too as it would give readers outside of the Philippines some context.

6. Methods, section 2.1. It's become increasingly important in research to state the specific study design rather than simply saying "qualitative study". The most common ones we see in applied health researh are qualitative descriptive or interpretive description (https://doi.org/10.1177%2F1744987119880234; https://doi.org/10.1111/medu.14380). Please identify your study design, state it here, and then provide 1-2 sentences on why you chose it for this particular study.

7. Methods. You should also following reporting guidelines (https://doi.org/10.1097/acm.0000000000000388), cite it in your methods, upload a copy of the table in the supplementary material indicating where you've addressed each point. It's extra work but it really helps advance good quality qualitative research and can help readers trust your findings.

8. Methods, Line 65. Why was the inclusion criteria >20 years old? That seems unusual to me as usually it's 18 years. Could you explain this in your methods section somewhere?

9. Methods. Since your study population is mostly women, I am also going to assume BHWs are generally also a primarily women dominated profession. Am I correct? If so, it would be helpful to highlight this to the reader in your introduction section that women traditionally go into this role.

10. Methods. Could you please upload the semi-structured interview guide as supplementary material and then cite it in section 2.3 somewhere stating that the guide is available?

11. Methods. Please include the following information in your methods section: (a) how long the interviews were, on average (e.g., M = 42 minutes) and/or the range (e.g., 20-55 minutes); (b) whether you used inductive or deductive coding; (c) whether you use latent or semantic coding; and (d) whether any triangulation methods were used during thematic analysis.

12. Results. Remove quotations from Lines 107-108 and 151-153 and instead leave them as regular text. You can number your themes here too.

13. Results. In section 3.2, you start numbering your themes from 1-4. Since you already have 3 themes listed in the previous section, I would suggest continuing the numbering from 4-7.

14. Results. It's also unusual to categorize themes into greater meta-themes (i.e., roles of BHWs and difficulties) because that isn't part of thematic analysis. I have no issues with it, but you should be transparent and state your reasoning for this in your methods under section 2.4. 

15. Please move Tables 2-3 to the supplementary material or integrate into the main text of the results.

16. Limitations. What about (i) your exclusion criteria of illiteracy and (ii) the gender-specific experiences of BHWs? That could have further affected validity of findings and you could state that in your limitations that male BHWs and illiterate BHWs could have different perspectives that may not have been adequately considered in the present study.

17. Discussion. After your limitations section, could you add a paragraph on future research directions? It could be a few sentences on what future investigations could focus on to improve health promotion in the Philippines and how specifically BHWs can deliver health information better to reduce inequities. I won't tell you exactly what to write here because you are the experts and should help guide the field forward.

18. Conclusion. This conclusion section could use a few more sentences. Particularly, I would suggest presenting the following information: (i) your most novel findings (e.g., Is this the first study of its kind? What does it offer that other studies don't offer?), (ii) recommendations on reducing barriers/difficulties/dangers of BHWs to conduct their jobs, (ii) recommendations on reducing health ineuities, and (iii) 1-2 suggestions on what future research could focus on.

19. Study title. I think your title could be revised to be a little shorter or catchier to attract more readership and citations. Here are two suggestions: (i) Community health workers' experiences in the Philippines: A qualitative study, and (ii) "It is hard but we enjoy our work": A qualitative study of Filipino community health workers' experiences

20. Overall. How do you feel about referring BHWs as community health workers throughout your paper? Since this journal is fit for a global audience, I think it may be better received if you used the community health workers as the predominant terminology after introducing the idea that BHWs are what they are called locally.

Round 2

Reviewer 1 Report

The conclusions are still too brief. Normally, the results from presented research are compared with other similar studies, shown in the literature review (too poor, still), international and national value added is stressed in conclusions. This is not done. The topic is close to the term "co-production". If the role of BHWs was not yet investigated in any paper (I do not have time capacity to check, but author/s state so), still some framing of the text into the co-production stream would imporve the text.

Reviewer 2 Report

Reading all the revisions made, one by one, I have seen that they have applied the improvements and the revisions. The current version of the manuscript is much richer and more rigorous. This study continues to have very local-contextual relevance due to the method used. The decision to accept the study is left to the editor.

Reviewer 3 Report

Thank you for your revised submission, addressing many of my comments, including the interview guide as supplementary material, adding more detail to your data analysis section, and expanding the limitations and conclusion sections. I feel your manuscript is much stronger now. One minor comment that I didn't feel was fully addressed:

1. I appreciated you citing the reporting guidelines by O'Brien et al. Many qualitative studies also have one sentence in their methods section stating something like "We followed the Standards for Reporting Qualitative Research guidelines for this study [18]." I would strongly suggest you do this. After doing so, double check the list of 21 standards to confirm you have indeed reported everything.

Round 3

Reviewer 1 Report

If editors are satisfied, I can accept.

Reviewer 3 Report

Thank you for your revised submission. I feel this is acceptable for publication in its current form.